# Start up: A French program to support patients with pulmonary arterial hypertension during the adjustment of prostanoids to the individualized optimal dose

Emmanuel Bergot[1]*, Marie Fertin[2], Mélanie Gallant Dewavrin[3], Grégoire Prévot[4], Julia Meijer[5], Coralie Hakibilen[5], Laurette Gruand[5], Sophie Gauthier[5], Andreea Todea[6], Thamer Boukerzaza[6], Xavier Jaïs[7]

1 Department of Pneumology, CHU, Caen, France, 2 Department of Cardiology and Vascular Disease, CHRU, Lille, France, 3 Association HTaPFrance, Beaune, France, 4 Department of Pneumology, CHU, Toulouse, France, 5 Patientys, Boulogne-Billancourt, France, 6 Department of Medical Affairs, Janssen-Cilag, Issy-les-Moulineaux, France, 7 Department of Pneumology, University Paris-Sud, Le Kremlin-Bicêtre, France

* bergot-e@chu-caen.fr

## Abstract

The Start Up program was initiated in patients with pulmonary arterial hypertension (PAH) with the objective to support them in taking their therapies targeting the prostacyclin pathway and determine the individualized optimal dose. This report focuses on selexipag, an oral prostacyclin receptor agonist that has been shown to delay the progression of PAH. Starting in 2016, patients who were prescribed selexipag for the treatment of PAH were offered participation in the Start Up program supported by the French PH network. During the dose adjustment phase, a nurse called the patients 3 days after drug initiation and after each dosage modification to provide support and to systematically grade and monitor adverse events (using Common Terminology Criteria for Adverse Events – CTCAE v4.0). The patients and the healthcare professionals rated their satisfaction with the program. Among the 406 patients who completed the program, 302 (74%) reached their individualized optimal dose for selexipag over a median period of 11 weeks. The reasons the individualized dose was not reached included treatment discontinuation (n = 78, 19%) and dropout for impaired communication skills (n = 8, 2%). Once the individualized optimal dose was reached, most patients had no, or mild (Grade 1) adverse events related to prostanoid use (49% and 45%, respectively). Both patients and healthcare professionals were satisfied with the Start Up program (scores of 9.1 ± 1.4 and 9.3 ± 0.7 on a 0–10 numerical scale, respectively). The proactive support and adverse event monitoring provided by the Start Up program during the early stages of treatment with selexipag was appreciated by patients and healthcare professionals and may contribute to improved compliance.

**Data availability statement:** Data are issued from the Start Up patient support program developed by Patientys. Patient gave their written consent for the use of their data, however they did not agree specifically for public availability of their unaggregated data. To meet PlosOne requirements, data can be available upon request at privacy@patientys.com.

**Funding:** This is a patient program managed by Patientys and financially supported by Janssen Cilag. Patientys contributed to program design, interpretation of the data, writing of the report, and decision to submit the paper for publication. Medical writing support was funded by Patientys. EB, MF, MGD for HTaPFrance, GP, and XJ report having been consultants for Patientys, and having received consulting fees, during the conduct of the study. AT and TB are employees of Janssen-Cilag. JM, CH, LG and SG are employees of Patientys. URL to sponsor's websites: https://www.janssen.com/france. There was no additional external funding received for this study.

**Competing interests:** The authors have declared that no competing interests exist.

## Introduction

Pulmonary arterial hypertension (PAH) is a rare disease caused by the remodeling of the pulmonary vasculature resulting in a progressive increase in pulmonary vascular resistance and, eventually, in right ventricular failure and premature death. Approved treatments include endothelin receptor antagonists (ERA), phosphodiesterase type 5 inhibitors (PDE5i), soluble guanylate cyclase stimulators, prostacyclin, prostacyclin analogs, and prostacyclin receptor agonists [1,2]. These therapies target the endothelin, nitric oxide, and prostacyclin pathways, which are involved in pulmonary vasoconstriction, and their combined use is supported by current treatment guidelines [3]. Oral dual combination therapy with an ERA and/or a PDE5i is recommended for most PAH patients who are at low or intermediate risk at diagnosis, while high risk patients should be treated with combination therapy, including intravenous prostacyclins [3–5].

The approval of selexipag, an oral prostacyclin receptor agonist in 2015, represented a major advance in the management of PAH. The GRIPHON study demonstrated that selexipag delayed disease progression in patients with PAH, independently of background therapy with an ERA and/or a PDE5i [6].

The Start Up program was initiated by the French PH network to provide a uniform management of prostacyclin-related adverse events occurring during prostanoid dose adjustment along with patient psychological support. These class-related events are more frequent during the initial dose adjustment phase, but they are usually transient and/or manageable with symptomatic treatment. As established with therapies that target the prostacyclin pathway, selexipag is individually adjusted to the highest dose at which the patient has manageable adverse events. The Start Up program was expected to support the ambulatory administration of selexipag by improving adverse event management and patient compliance to treatment during the critical dose adjustment period.

## Materials and methods

### Design

The Start Up program was designed to support patients with PAH (Group 1) during the adjustment of therapies targeting the prostacyclin pathway to the individualized optimal dose. Initiated by a healthcare nonprofit association, the Start Up patient support program is implemented by Patientys, under the supervision of a scientific medical committee composed of French physicians from French PAH network and a patient representative from the patient organization HTaPFrance (HyperTension arterielle Pulmonaire France). The program benefits from the institutional support of Janssen Cilag France.

A dedicated PAH nurse, employed by Patientys, the external patient service in charge of the Start Up program, called the patient 3 days after drug initiation and 3 days after each dosage modification to provide listening support and drive the patient interview in order to systematically grade and monitor adverse events related to prostacyclin use (S1 Fig). A report was systematically sent to the treating physician by e-mail. Satisfaction with the program was rated by the patient and the physician. As

a result of this survey conducted in 2018, after reaching their individualized optimal dose, a one year follow-up has been added to the Start Up program. Therefore, among the 302 patients that reached their individualized dose, 160 patients benefited from the titration phase and the additional follow-up (sub-population 2 – P2) while 142 patients benefited from the titration phase only (sub-population 1 - P1) (Fig 1). During this year following program completion, the patient treatment dose was documented by a phone call at Month 1, 3, 6, 9 and 12.

## Patients

Eligible patients were those prescribed selexipag for the treatment of PAH (Group 1), mainly via the French PH network, which includes the French reference center for PH (Le Kremlin-Bicêtre, France) and 28 regional reference or competence center PH centers widespread across France. Written informed consent was obtained for each patient participating in the program.

## Selexipag dose adjustment

Patients were titrated to their individualized optimal dose, which could range from 200 µg twice daily to 1,600 µg twice daily, as per routine practice and treating physician judgment [6]. Selexipag was initiated at a dose of 200 µg twice daily and was up-titrated, usually at weekly intervals, in twice-daily increments of 200 µg until the 1,600 µg maximum dose was reached, or until unmanageable adverse events related to prostacyclin use occurred. In that case, the dose was reduced to the previous dose level, which was defined as the patient's individualized optimal dose. A second attempt to continue up-titration to the individualized optimal dose up to a maximum dose of 1,600 µg twice daily could be considered based on

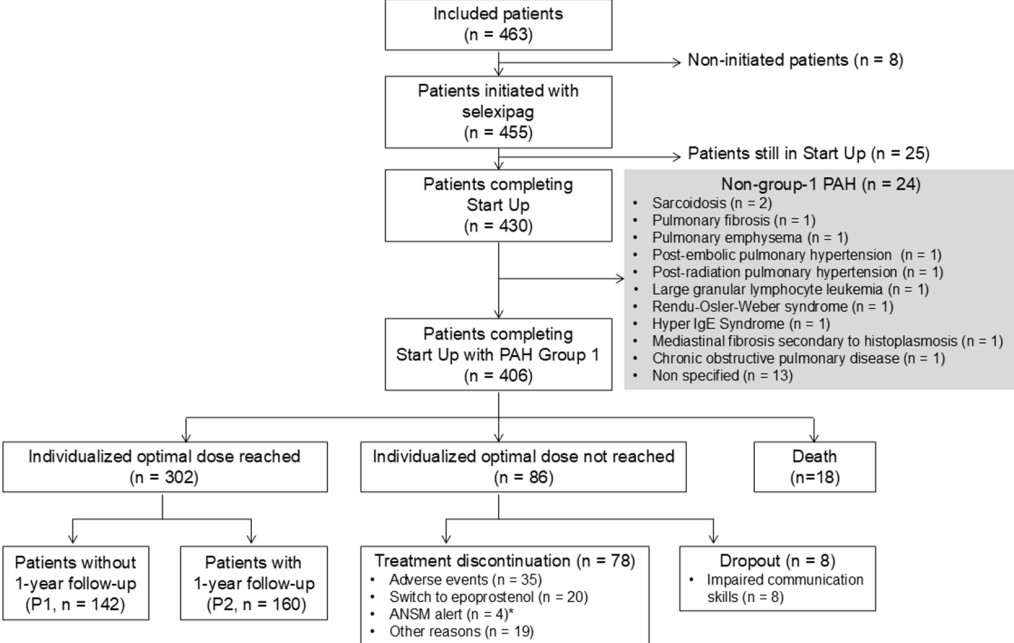

**Fig 1. Flow chart.** *A 3-month alert from the ANSM (*Agence Nationale de Sécurité du Médicament et des Produits de Santé* - French drug regulatory agency) resulted in some patients/physicians deciding to discontinue treatment. The alert resolved in April 2017 with no additional impact on participation (https://ansm.sante.fr/S-informer/Informations-de-securite-Lettres-aux-professionnels-de-sante/Levee-de-la-mesure-de-precaution-concenant-Uptra-vi-selexipag-Lettre-aux-professionnels-de-sante). †The one-year follow-up was defined as complete for the patients who responded to the 5 programmed phone calls (at Month 1, 3, 6, 9, and 12), it was partial for those who responded to at least 1 phone call during the one-year follow-up, and in progress for those who had not completed the one-year follow-up at the time of data cutoff.

the treating physician judgment. The individualized optimal dose reached is categorized in the analysis below in 3 categories: low (200–400 ug), medium (600–1000 ug), and high (1400–1600 ug).

### Grading and management of prostacyclin-related adverse events

Adverse events that are typically related to prostacyclin use, including headache, diarrhea, myalgia, jaw pain, nausea, flushing, vomiting, and arthralgia, were monitored and graded by the PAH nurse of the Start Up program. The severity of adverse events was graded using the Common Terminology Criteria for Adverse Events (CTCAE Version 4.0). The CTCAE displays Grades 1–5 with unique clinical descriptions of severity for each adverse event: Grade 1 events are mild (asymptomatic or mild symptoms; clinical or diagnostic observations only; intervention not indicated), Grade 2 events are moderate (minimal, local or noninvasive intervention indicated; limiting age-appropriate instrumental Activities of Daily Living (ADL)), Grade 3 events are severe (medically significant but not immediately life-threatening; hospitalization or prolongation of hospitalization indicated; disabling; limiting self-care ADL), Grade 4 events have life-threatening consequences, and Grade 5 events are related to death. The event 'jaw pain', which is not displayed in the CTCAE was assessed as mild (Grade 1), moderate (Grade 2) or severe (Grade 3). Three days after drug initiation and 3 days after each dosage modification, the nurse called the patient to document adverse events. Adverse events with Grade ≥ 2 were immediately reported by e-mail to the treating physician and patients were advised about symptomatic treatment and/or dosage adjustment. A follow-up call was performed 3 days later and persistent adverse events with Grade ≥ 2 were signaled again by e-mail to the physician for appropriate management. The last call was performed 7 days after the patient had reached the individualized optimal dose. At that time, the burden of adverse events was defined for each patient as the sum of the severity grades reported for each ongoing prostacyclin-related adverse event.

### Satisfaction questionnaires

Starting January 2018, patients and healthcare professionals were asked to complete a questionnaire after the last call of the dose adjustment phase to assess their level of satisfaction regarding the program. Using a 0 (totally unsatisfied) to 10 (very satisfied) numerical scale, patients rated global satisfaction, program helpfulness (regarding dose adjustment, adverse event management, treatment compliance, patient-physician relationship, treatment intake and day-to-day disease management). In addition to grading their satisfaction, patients were also asked to rank the benefits they derived from the program among 7 proposals (help with adverse event management, treatment acceptance, dose adjustment, and relationship with physician, listening, psychological support and exchange on the disease with the nurse). Patients also evaluated the frequency of the nurse calls (not enough, appropriate, too often) and the usefulness of the program (essential, useful, moderately useful, not useful). Healthcare professionals similarly use a 0–10 numerical scale to rate global satisfaction, specific program features (e-mail communication/alerts, program content, and relevance of transmitted information), and program helpfulness regarding patient-physician relationship. Healthcare professionals were asked whether the program was a time saver for themselves and for the medical team, whether they always proposed the program to their patients and whether they had patients who refused to participate.

### Statistical analysis

Data were summarized using descriptive statistics reporting medians with ranges, means with standard deviations, or numbers with percentages. Data was anonymized for analysis and accessed for research purposes from the 1st of November 2022. Authors did not have access to information that could identify individual participants during or after data collection.

Data collection and analysis were carried out in accordance with french Reference Methodology MR004 (#2235657), published by the French Data Protection Authority (Commission Nationale de l'Informatique et des libertés).

## Results

### Study population

From September 2016 to October 2022, 463 patients have been enrolled in the Start Up program and 406 patients with PAH (group 1) have completed the program (Fig 1). Among these patients, 302 (74%) reached their individualized optimal dose for selexipag, 86 (21%) did not, and 18 (4%) died (with no information about the cause for the majority). The reasons the individualized optimal dose was not reached included treatment discontinuation (n = 78, 19%) and dropout for impaired communication skills (n = 8, 2%).

During the titration phase of the program, patients (n = 302) received a mean number of 18 ± 10 calls from the PAH nurse of a mean duration of 13 ± 7 minutes each, corresponding to an overall mean duration of 3 hours and 50 minutes. This first period of treatment titration was dedicated to help patients determining their individualized optimal dose, managing adverse events, and maintaining treatment adherence. On the 302 patients that reached their optimal dose, 160 patients (P2) had the possibility to receive follow-up calls (53%). This opportunity given to patients is an evolution of the Start-Up program introduced from March 2019 regarding the satisfaction survey answers. This second period of follow-up consisted in regular call from the Start Up nurse (at Month 1, 3, 6, 9, and 12). Conversations with patients were devoted to empathetic listening and reassuring, reminding the correct drug use, and relaying medical team messages to patients. A systematic report was shared to the treating physician providing insights about the patients in between hospital appointments. It also allows if necessary to start again a new titration. This second sub-population of patients (P2) received a mean number of 20 ± 11 calls for titration and a mean number of 4 ± 2 follow-up calls of a mean duration of 18 ± 7 minutes. As a comparison, patients that only received call during the titration phase (P1) had an average overall call time of 3 hours and 50 minutes, while patients of P2 have had an average call time of 6 hours and 13 minutes.

### Baseline demographic and disease characteristics

Patients with PAH enrolled in the Start Up program had a mean age of 57.8 ± 16.6 years and included a large proportion (68%) of women (Table 1). Most patients were in World Health Organization (WHO) functional class II-III (88%) and were already receiving PAH-specific dual therapy with an ERA and a PDE5i (92%) at time of selexipag initiation.

### Individualized optimal dose

The individualized optimal dose was reached in 302 out of 406 (74%) patients over a median period of 11 weeks (range: 3.1–117 weeks) with 168 (56%) patients reaching their individualized dose in 12 weeks or less (Table 2). The median dose was 1400 µg twice daily. The optimal dose was high (1200–1600 µg twice daily) for 188 (62%) patients, medium (600–1000 µg twice daily) for 91 (30%) patients, and low (200–400 µg twice daily) for 24 (8%) patients (Fig 2). The dose could be up-titrated step-by-step (200 µg twice daily increment) in 176 (57%) patients, although the up-titration was performed at a slower pace than weekly in 90 (28%) patients (Fig 3). The identification of the individualized optimal dose required one or several dose decrease(s) in 71 (23%) patients and 33 (11%) patients, respectively. A dose decrease was followed by a tentative increase in 5 (2%) patients (Fig 3). Seventeen (6%) patients did not fit any of the above patterns.

### Occurrence and management of adverse events related to prostacyclin use

During the dose-adjustment phase, adverse events related to prostacyclin use were reported by most patients (97%). They most frequently included headache (77%), diarrhea (71%), myalgia (59%), nausea (48%), and jaw pain (44%), (Table 3). Overall, 1545 events were reported: 41% were Grade 1, 34% were Grade 2, and 25% were Grade 3.

At the end of the dose adjustment phase, most patients had no or mild (Grade 1) prostacyclin-related adverse events (49% and 45%, respectively). Fifteen (5%) patients had a Grade 2 event, and two (1%) patients had a Grade 3 event (for one patient, the event was jaw pain and the optimal dose for this patient was 400 µg twice daily, and for the other patient

**Table 1. Baseline demographic and disease characteristics of patients completing the Start Up program.**

| Characteristics | Patients who reached individualized optimal dose (n = 302) | Patients who did not reach individualized optimal dose or died (n = 104) | All patients (n = 406) |
|---|---|---|---|
| Age, years (mean ± STD) | 56.9 ± 17.1 | 59.2 ± 14.6 | 57.8 ± 16.6 |
| Gender – women, n (%) | 201 (67%) | 74 (71%) | 275 (68%) |
| PAH (Group 1) classification, n (%) | | | |
| Idiopathic | 149 (49%) | 45 (43%) | 194 (48%) |
| Heritable | 31 (10%) | 7 (7%) | 38 (9%) |
| Associated with connective tissue disease | 64 (21%) | 28 (27%) | 92 (23%) |
| Associated with corrected congenital shunts | 31 (10%) | 13 (13%) | 44 (11) |
| Associated with drug or toxin exposure | 16 (5%) | 5 (5%) | 21 (5%) |
| Associated with HIV infection | 4 (1%) | – | 4 (1%) |
| Other | 6 (2%) | 6 (6%) | 12 (3%) |
| WHO functional class, n (%) | | | |
| II | 82 (27%) | 21 (20%) | 103 (26%) |
| III | 191 (63%) | 65 (62%) | 256 (63%) |
| IV | 15 (6%) | 10 (10%) | 25 (6%) |
| Unknown | 14 (4%) | 8 (8%) | 22 (5%) |
| PAH medications, n (%) | | | |
| ERA | 9 (3%) | 4 (4%) | 13 (3%) |
| PDE5i | 15 (5%) | 4 (4%) | 19 (5%) |
| Dual therapy (ERA and PDE5i) | 278 (92%) | 95 (91%) | 373 (92%) |
| Unknown | – | 1 (1%) | 1 (0%) |

ERA: Endothelin receptor antagonist HIV: human immunodeficiency virus; PAH: pulmonary arterial hypertension; PDE5i: Phosphodiesterase type 5 inhibitor; STD: standard deviation; WHO: World Health Organization.

**Table 2. Patient distribution according to dose and time to reach individualized optimal dose (n = 302).**

| Dose range (twice daily) | Patients n (%) | |
|---|---|---|
| | Time to reach optimal dose | |
| | ≤12 weeks* | >12 weeks* |
| Low dose (200–400 µg) (n = 24) | 15 (63%) | 9 (38%) |
| Medium dose (600–1000 µg) (n = 91) | 52 (57%) | 39 (43%) |
| High dose (1200–1600 µg) (n = 187) | 100 (63%) | 87 (47%) |
| All doses (n = 302) | 167 (55%) | 135 (45%) |

*A 12-weeks threshold was chosen to match the duration of the dose adjustment period in the licensing phase III GRIPHON trial [6]

the events were jaw pain and headache for an optimal dose of 600 µg twice daily). The burden of adverse events defined as the sum of the prostacyclin-related adverse event severity levels per patient was 2 or below in most (86%) patients (Table 4).

The severity of adverse events was graded according to the Common Terminology Criteria for Adverse Events (CTCAE) Version 4.0, which displays Grade 1 (mild AE) through Grade 5 (death related to AE). Grade 4 and 5 prostacyclin-related adverse events did not occur during the Start Up program.

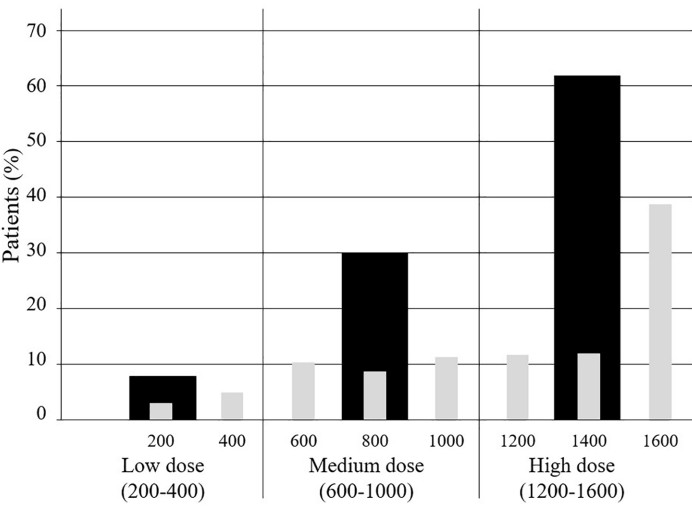

**Fig 2. Selexipag individualized optimal dose (n = 302).** The grey bars represent the percentage of patients for each dose. The black bars represent the percentage of patients for each category of dose (Low, Medium or High dose).

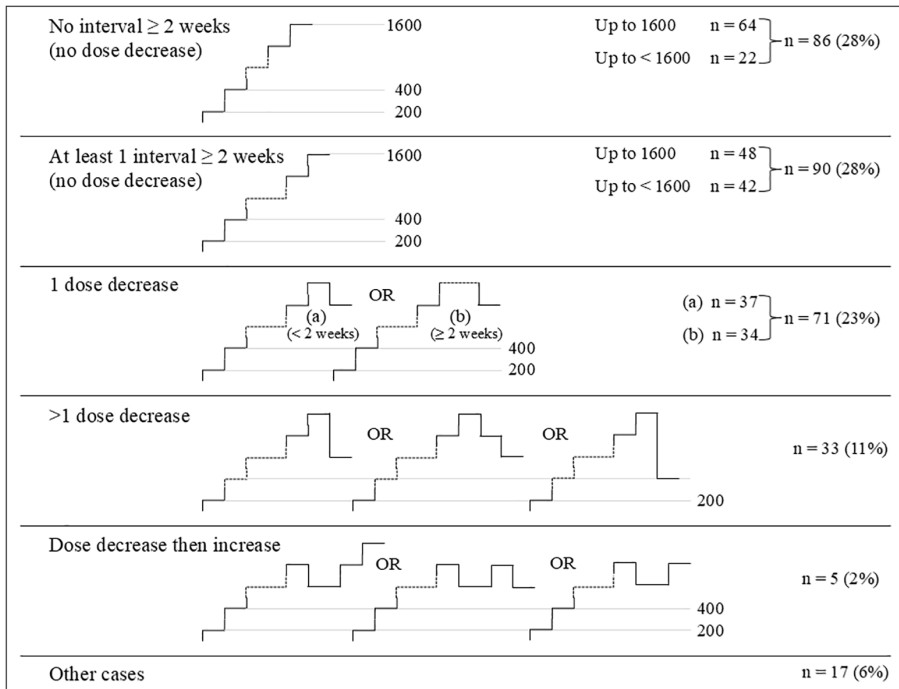

**Fig 3. Selexipag dose adjustment patterns in patients reaching their individualized optimal dose (n = 302).**

## Program dropouts – treatment discontinuations- deaths

During the selexipag adjustment phase, 8 (2%) patients dropped out of the program after it became clear their impaired communication skills did not allow for a phone exchange. In addition, treatment was discontinued and program was

**Table 3. Patients with prostacyclin-related adverse events during the dose-adjustment phase (n = 406).**

| Prostacyclin-related adverse event | Patients with adverse events | | | |
|---|---|---|---|---|
| | Adverse event severity level[a] | | | |
| | Any grade AE n (%) | Grade 1 (mild) n | Grade 2 (moderate) n | Grade 3 (severe) n |
| Headache | 314 (77%) | 85 | 107 | 122 |
| Diarrhea | 288 (71%) | 137 | 95 | 56 |
| Myalgia | 240 (59%) | 59 | 90 | 91 |
| Nausea | 194 (48%) | 74 | 100 | 20 |
| Jaw pain | 179 (44%) | 98 | 36 | 45 |
| Flushing | 145 (36%) | 70 | 47 | 28 |
| Vomiting | 99 (24%) | 80 | 17 | 2 |
| Arthralgia | 86 (21%) | 31 | 31 | 24 |

[a]The severity of adverse events was graded according to the Common Terminology Criteria for Adverse Events (CTCAE) Version 4.0, which displays Grade 1 (mild AE) through Grade 5 (death related to AE). Grade 4 and 5 prostacyclin-related adverse events did not occur during the Start Up program.

**Table 4. The burden of prostacyclin-related adverse events[a] at the end of the dose-adjustment phase in patients reaching their individualized optimal dose (n = 302).**

| Adverse event burden[b] | Patients, n (%) |
|---|---|
| None | 137 (45%) |
| 1 | 69 (23%) |
| 2 | 54 (18%) |
| 3 | 27 (9%) |
| 4 | 7 (2%) |
| 5–7 | 8 (3%) |
| > 7 | 0 |

[a]Prostacyclin-related adverse events included headache, diarrhea, myalgia, nausea, jaw pain, flushing, vomiting, and arthralgia.

[b]Defined as the sum of the prostacyclin-related adverse event severity levels per patient

stopped in 78 (19%) patients. For 4 of these patients, the decision was taken following a 3 month salert on selexipag from the French drug regulatory agency (ANSM – Agence Nationale de Sécurité du Médicament et des Produits de Santé). The alert was resolve in April 2017 after examination from the European Committee for risk evaluation and pharmacovigilance stating that data did not suggest any excess mortality with selexipag; no additional impact on participation was observed [7]. For 74 (18%) patients, the physician decided treatment was to be stopped for adverse event (n = 35), need to switch the patient to intravenous epoprostenol (n = 20) or other reasons (n = 19). Eighteen (4%) patients died.

## Satisfaction questionnaires from patients and healthcare professionals

Of the 302 patients reaching the optimal dose, 249 patients received the satisfaction questionnaire and 86 (35%) completed it. Participants rated their global satisfaction as 9.1 ± 1.4 on the 0–10 numerical scale (Fig 4A). In addition, when asked which was the top-ranking benefit they derived from the program, they quoted help with dose adjustment (20%), listening (19%), help with treatment acceptance (15%) and adverse event management (14%). The frequency of calls from the nurse was deemed appropriate (97%) and the program was considered as 'essential' or 'useful' by 53% and 42% of the participants, respectively.

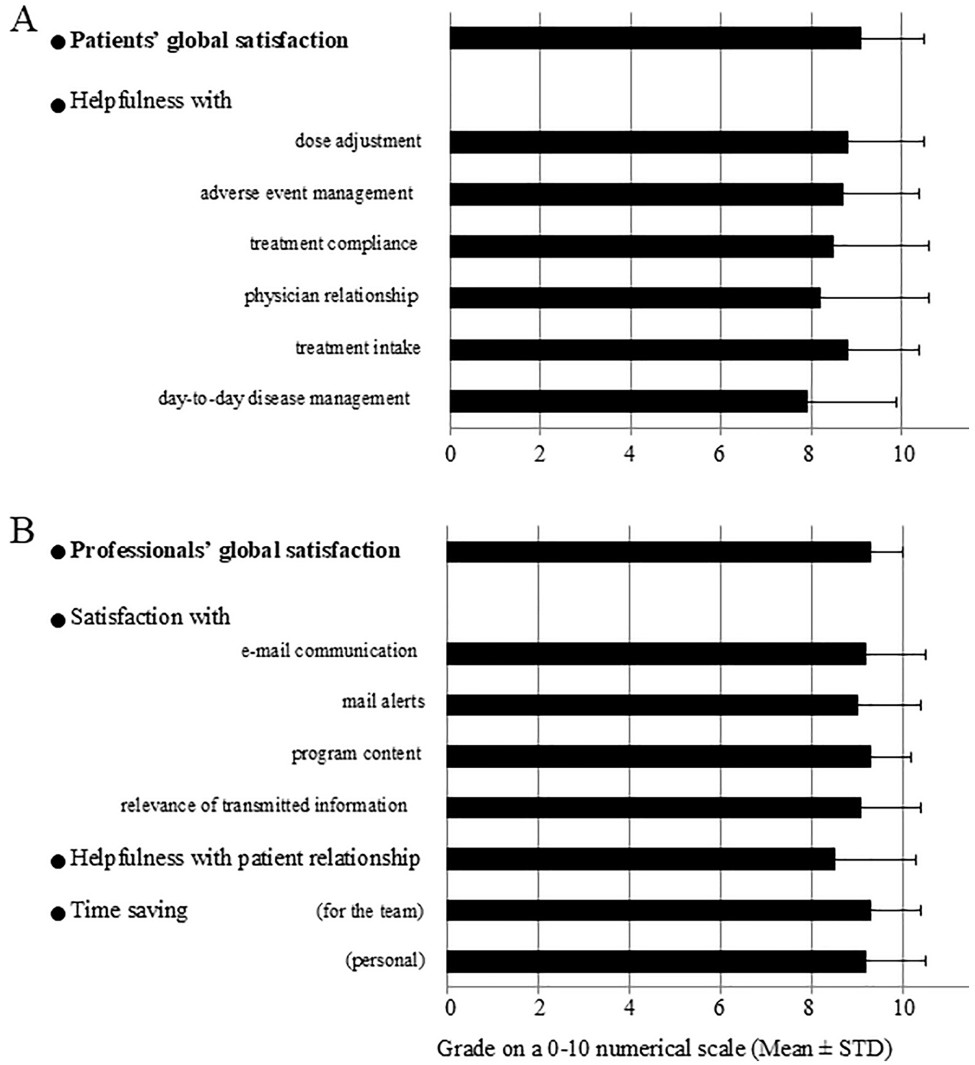

**Fig 4. Patients' and professionals' satisfaction with the program.** (A) for the patients (86 filled questionnaires out of 249 sent), (B) for the health-care professionals (51 filled questionnaires). Questionnaires were sent to patients and healthcare professionals starting January 2018. STD: standard deviation.

The 51 healthcare professionals (39 physicians, 1 pharmacist, 8 nurses, 3 unspecified) who completed the satisfaction questionnaire rated their global satisfaction as 9.3±0.7 on the 1–10 numerical scale (Fig 4B). The program was associated with time saving for the team and for the surveyed professional (rated as 9.3±1.1 and 9.2±1.3, respectively). Most healthcare professionals reported that they always proposed the program to their patients (90%) and that patients never refused to participate (94%).

## One-year follow-up after program completion

During the one-year follow-up after program completion, 65 patients responded to the 5 programmed phone calls at Month 1, 3, 6, 9, and 12 (complete one-year follow-up) and 69 responded to at least 1 phone call (partial one-year follow-up). Among these 134 patients, 23 (17%) required a dose modification during the one-year follow-up (dose increase for 5% patients, decrease for 11%, increase followed by decrease for 1%). Among the 65 patients with complete one-year

follow-up, a dose change was required for about 20% patients during the first quarter and for about 10% patients during each of the 3 subsequent quarters (Table 5).

## Discussion

In this real-life follow-up of a large cohort of patients receiving ambulatory selexipag for the treatment of PAH, the Start Up program was set up to help the patients with drug adjustment to the optimal dose, a process that is common to all prostanoid therapies but can be worrisome for the patient. Individualized support and proactive adverse event monitoring during the early stages of treatment were appreciated by both the patients and the healthcare professionals, suggesting the Start Up program may contribute to improving early treatment compliance.

The French PH network, which designed the Start Up program, was initially set up by the French reference center for PH (Le Kremlin-Bicêtre, France) and now includes 32 regional centers widespread across France. With the Start Up program, the French PH network aimed to improve patients' confidence in their treatment by having a PAH nurse proactively contacting them during the early stages of treatment, when they are more likely to be experiencing class-specific adverse events such as headache, diarrhea, nausea, or jaw pain.

With over two thirds of the patients in class III or IV, the Start Up program included patients with very severe PAH compared with the licensing phase III GRIPHON trial [6], the US SPHERE registry [8], or the ongoing EXPOSURE cohort study conducted in Europe and Canada [9]. In addition, the real-life patients enrolled in Start Up, SPHERE, and EXPOSURE were older than those participating in the GRIPHON trial. This may explain the elevated rates observed for death (4%) and switch to epoprostenol (5%) over the short dose adjustment period in the Start Up program. Frequent discontinuation due to a prostacyclin-related adverse event was expected during that period, yet, despite age and disease severity, relatively few patients (9%) permanently stopped their treatment because of an unmanaged adverse event. As a comparison, among real-world data studies, the discontinuation rate during the follow-up period due to adverse events varies from 20% in the EXPOSURE study and 22% in the SPHERE registry study up to 32% in the GRIPHON open label study [10–12]. However, adverse events occur more frequently during the dose-adjustment phase [6]. We may therefore speculate for higher treatment discontinuation during the titration phase of these studies as compared to the rate observed in the Start Up program, supporting that close patient monitoring and a slower treatment titration in regard to treatment tolerability may participate in less treatment discontinuation.

In line with current approaches emphasizing the importance of individualized support and patient education in the optimal therapeutic management of PAH patients [13,14], the Start Up nurse played a crucial role in helping the patient with specific features of the selexipag treatment including dose adjustment, class-specific adverse event identification and prompt reporting to the treating physician, while managing patients' expectations and questioning on their treatment and/or disease. During the dose adjustment phase, the nurse spent on average 4 hours communicating with each patient, but one patient benefited from 16 hours of individual support. While treatment uptitration is sometimes fully handled by the hospital teams, we show how much time this may represent, while it may have resulted in substantial time saving for the treating physician and its healthcare team. The strength of this program also relies on the possibility

**Table 5. Patients with dose modification in the year following program completion (for those patients with complete follow-up*, n = 65).**

|  | Patients reporting dose modification between | | | | |
|  | Month 0–1 | Month 1–3 | Month 3–6 | Month 6–9 | Month 9–12 |
|---|---|---|---|---|---|
| Dose increase | 5 (8%) | 3 (5%) | 1 (2%) | 0 | 1 (2%) |
| Dose decrease | 2 (3%) | 4 (6%) | 6 (9%) | 6 (9%) | 5 (8%) |
| No change | 58 (89%) | 58 (89%) | 58 (89%) | 59 (91%) | 59 (91%) |

*65 patients responded to the 5 programmed phone calls during the one-year follow-up after program completion and were considered to have a complete follow-up.

for the PAH nurse to adapt and complete the phone calls timeline. Indeed, during the titration phase, if an adverse event was detected during a phone call, an additional phone call was proposed 3 days after to the patient. Therefore the nurse could help the patient managing adverse events, reassure him and reinforce messages of the medical team if needed. This coordination function of the Start Up nurse was specifically highly useful for PAH centers without hospital coordination resources. The helpfulness of the program was assessed favorably by the patients and healthcare professionals in the satisfaction questionnaires, resulting in the program being prolonged for an additional year after the optimal dose was reached. This additional follow-up indicated that dose modification can be necessary after the patient has reached the optimal dose, which supports the usefulness of a program such as Start Up to help the patient with long-term treatment tolerance and compliance.

The Start Up patients benefited from an individualized dose adjustment scheme that was tailored to their drug tolerability profile by their treating physicians, who were informed in real time on the development and the severity grade of adverse events. The events were objectively assessed and graded using the CTCAE (Version 4.0) allowing for systematic and efficient management. As a result, the dose adjustment was slower in some patients (45%) than the maximum 12-week duration mandated by the GRIPHON protocol [6], but a trend was observed for higher optimal doses achieved when compared with the GRIPHON study [6] or with the US SPHERE registry [8], which investigated the real-world dosing adjustment of selexipag. In the GRIPHON study the median maintenance dose was 1000 µg twice daily, achieved in less than 12 weeks in all patients. In the SPHERE registry, a median maintenance dose of 1200 µg twice daily was reached over a median period of 8.1 weeks. For comparison, in the Start Up program, the median optimal dose was 1400 µg twice daily achieved over a median period of 11 weeks. A recent real-life selexipag study in 26 patients with pulmonary hypertension enrolled in a support program similar to Start Up in Germany also reported high optimal doses (median of 1450 µg twice daily) reached after a slow adjustment period (median of 12.9 weeks) [15]. The program-supported studies may offer a more flexible dose adjustment regimen, which, along with patient education and proactive support, may encourage the patients to persevere with their treatment routine beyond the first weeks and thus facilitate the adjustment to optimal dosages. Among the 302 patients of the Start Up program reaching their individualized optimal dose, 180 (62%) patients either reached the 1600 µg maximum dose allowed (n = 112) or were down-titrated to the previous dose level for tolerability reason (n = 71), as recommended in the prescribing information of selexipag [16]. However, additional patterns of dose adjustment could be seen in this real-life study such as dose maintenance below the maximum dose allowed with no decrease to the previous dose level (n = 64, 21%), or multiple dose decrease (n = 33, 11%), or tentative increase after a decrease (n = 5, 1%). Lombardi *et al.* [17] have suggested increasing only one of the two daily doses at a time at weekly intervals in patients with tolerability issues. Emerging experience seems to support a slow, cautious dose adjustment of selexipag to optimize patient compliance and therapeutic benefit.

As expected from the GRIPHON trial, adverse events typically related to prostacyclin therapy, especially headache diarrhea, myalgia, jaw pain and nausea, were frequently reported during the dose adjustment of selexipag. These events significantly impact the quality of life of the patients, but management strategies are available to help patients stay on treatment and receive the maximum benefit from their prostacyclin therapy [18]. The frequency of adverse events was high in the Start Up program and only ten patients (2%) did not experience a prostacyclin-related adverse event compared with 13% in the GRIPHON trial. It is possible that the multilevel CTCAE grading of adverse events allowing for the recording of mild and minimal symptoms, as well as the frequent calls of the Start Up nurse inquiring about specific events may have resulted in a lower threshold for the reporting of adverse events.

## Limitations

The first limitation of the current study lies in its design. The first intent of the Start Up program was to support patients and their medical team, therefore, it does not allow for a comparison with patients who did not benefit from the patient support program, nor is there a direct statistical analysis with existing studies.

Our analysis of the satisfaction of the patients and the health care professionals is limited by the late implementation of this evaluation, resulting in early participants enrolled before December 2017 not filling the questionnaires. The low rate of response to the survey and the fact that it was only proposed to those patients who reached their optimal dose suggest that responses may not be representative of the overall population satisfaction. In addition, the risk of non-response bias could have introduced an over-estimation of patient satisfaction. The questionnaires were sent to physicians but forwarded to other health care professionals with no control over it, which limited our rate of response. The responses of other healthcare professionals allow for better representation of people involved in the patient's care pathway. All these issues have been addressed to improve future reporting, but current results should be interpreted with caution.

## Conclusion

The Start Up program has been designed to support patients with PAH (Group 1) during the adjustment of therapies targeting the prostacyclin pathway to the individualized optimal dose. In this real-life follow-up of a large cohort patients receiving ambulatory selexipag, both the healthcare professionals and the patients reported their satisfaction with the program. By providing proactive support and adverse event monitoring personalized to the patient during the critical dose adjustment phase, the Start Up program may have encouraged the patients to comply with their treatment and possibly helped them reach their individualized optimal dose.

## Supporting information

**S1 Fig. Standard patient management scheme for the Start Up program.**
(TIF)

## Acknowledgments

The authors thank the PAH nurse team of the Start Up program, for her commitment to helping the patients, the patient organization HTaPFrance for their support to patients and participation in the program design and S. I. Ertel (Sundgau Medical Writers, France) for editorial assistance.

## Author contributions

**Conceptualization:** Emmanuel Bergot, Xavier Jaïs.

**Formal analysis:** Julia Meijer, Coralie Hakibilen, Laurette Gruand, Sophie Gauthier.

**Funding acquisition:** Andreea Todea, Thamer Boukerzaza.

**Investigation:** Emmanuel Bergot, Marie Fertin, Mélanie Gallant Dewavrin, Grégoire Prévot, Julia Meijer, Coralie Hakibilen, Laurette Gruand, Sophie Gauthier, Xavier Jaïs.

**Project administration:** Julia Meijer, Coralie Hakibilen, Laurette Gruand, Sophie Gauthier.

**Supervision:** Andreea Todea, Thamer Boukerzaza.

**Validation:** Thamer Boukerzaza.

**Visualization:** Julia Meijer, Coralie Hakibilen, Laurette Gruand, Sophie Gauthier, Andreea Todea.

**Writing – original draft:** Emmanuel Bergot.

**Writing – review & editing:** Emmanuel Bergot, Marie Fertin, Mélanie Gallant Dewavrin, Grégoire Prévot, Julia Meijer, Coralie Hakibilen, Laurette Gruand, Sophie Gauthier, Andreea Todea, Thamer Boukerzaza, Xavier Jaïs.

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
