## [Decision Letter · Decision Letter 0]

5 May 2025

PONE-D-25-06416Start Up: A French program to support patients with pulmonary arterial hypertension during the adjustment of prostanoids to the individualized optimal dosePLOS ONE

Dear Dr. Bergot,

Thank you for submitting your manuscript to PLOS ONE. After careful consideration, we feel that it has merit but does not fully meet PLOS ONE’s publication criteria as it currently stands. Therefore, we invite you to submit a revised version of the manuscript that addresses the points raised during the review process.

We look forward to receiving your revised manuscript.

Kind regards,

James P. Maloney, MD

Academic Editor

PLOS ONE

**Journal Requirements:**

1. When submitting your revision, we need you to address these additional requirements. Please ensure that your manuscript meets PLOS ONE's style requirements, including those for file naming. The PLOS ONE style templates can be found at https://journals.plos.org/plosone/s/file?id=wjVg/PLOSOne_formatting_sample_main_body.pdf and https://journals.plos.org/plosone/s/file?id=ba62/PLOSOne_formatting_sample_title_authors_affiliations.pdf 2. Thank you for stating in your Funding Statement: This is a patient program managed by Patientys and financially supported by Janssen Cilag. Patientys contributed to program design, interpretation of the data, writing of thereport, and decision to submit the paper for publication. Medical writing support wasfunded by Patientys.EB, MF, MGD for HTaPFrance, GP, and XJ report having been consultants forPatientys, and having received consulting fees, during the conduct of the study.AT and TB are employees of Janssen-Cilag. JM, CH, LG and SG are employees ofPatientys.URL to sponsor's websites: https://www.janssen.com/france Please provide an amended statement that declares *all* the funding or sources of support (whether external or internal to your organization) received during this study, as detailed online in our guide for authors at http://journals.plos.org/plosone/s/submit-now.  Please also include the statement “There was no additional external funding received for this study.” in your updated Funding Statement. Please include your amended Funding Statement within your cover letter. We will change the online submission form on your behalf. 3. For studies involving third-party data, we encourage authors to share any data specific to their analyses that they can legally distribute. PLOS recognizes, however, that authors may be using third-party data they do not have the rights to share. When third-party data cannot be publicly shared, authors must provide all information necessary for interested researchers to apply to gain access to the data. (https://journals.plos.org/plosone/s/data-availability#loc-acceptable-data-access-restrictions)  For any third-party data that the authors cannot legally distribute, they should include the following information in their Data Availability Statement upon submission:a) A description of the data set and the third-party sourceb) If applicable, verification of permission to use the data setc) Confirmation of whether the authors received any special privileges in accessing the data that other researchers would not haved) All necessary contact information others would need to apply to gain access to the data 4. We note that you have indicated that there are restrictions to data sharing for this study. For studies involving human research participant data or other sensitive data, we encourage authors to share de-identified or anonymized data. However, when data cannot be publicly shared for ethical reasons, we allow authors to make their data sets available upon request. For information on unacceptable data access restrictions, please see http://journals.plos.org/plosone/s/data-availability#loc-unacceptable-data-access-restrictions.  Before we proceed with your manuscript, please address the following prompts: a) If there are ethical or legal restrictions on sharing a de-identified data set, please explain them in detail (e.g., data contain potentially identifying or sensitive patient information, data are owned by a third-party organization, etc.) and who has imposed them (e.g., a Research Ethics Committee or Institutional Review Board, etc.). Please also provide contact information for a data access committee, ethics committee, or other institutional body to which data requests may be sent. b) If there are no restrictions, please upload the minimal anonymized data set necessary to replicate your study findings to a stable, public repository and provide us with the relevant URLs, DOIs, or accession numbers. Please see http://www.bmj.com/content/340/bmj.c181.long for guidelines on how to de-identify and prepare clinical data for publication. For a list of recommended repositories, please see https://journals.plos.org/plosone/s/recommended-repositories. You also have the option of uploading the data as Supporting Information files, but we would recommend depositing data directly to a data repository if possible. Please update your Data Availability statement in the submission form accordingly. 5. Please include your full ethics statement in the ‘Methods’ section of your manuscript file. In your statement, please include the full name of the IRB or ethics committee who approved or waived your study, as well as whether or not you obtained informed written or verbal consent. If consent was waived for your study, please include this information in your statement as well.

**Additional Editor Comments:**

The manuscript offers useful information to  PH centers that do not have an active selexipag titration protocol, and offers important insights on dose maximization. Please address all reviewer comments

Reviewers' comments:

Reviewer's Responses to Questions

**Comments to the Author**

1. Is the manuscript technically sound, and do the data support the conclusions?

Reviewer #1: Partly

Reviewer #2: Yes

2. Has the statistical analysis been performed appropriately and rigorously? 

Reviewer #1: N/A

Reviewer #2: Yes

3. Have the authors made all data underlying the findings in their manuscript fully available?

Reviewer #1: No

Reviewer #2: No

4. Is the manuscript presented in an intelligible fashion and written in standard English?

Reviewer #1: Yes

Reviewer #2: Yes

5. Review Comments to the Author

**Reviewer #1:** Prostacyclin initiation and titration can be difficult and time consuming, so it is nice to see clinical study in this area. In the current paper, Bergot et al propose a “start-up program” during which patients would be supported by an external nurse during initiation. While there are a number of more rigorous questions that might have been attempted, the current paper is a bit superficial – there is no direct comparison group from the same patient population(s), either concurrent or historical. And the goal is nebulous – perhaps patient satisfaction, though if so the number completing surveys was low. Or perhaps number reaching their “maximally tolerated dose”, but this is not objectively defined and perhaps really can’t be.

The study does provide some observational data describing demographics, type of AEs, how long the outside nurse spent on the phone, and what dose they reached, patient and physician satisfaction - but none of this other than the phone call durations and satisfaction rates are novel as similar information is present in RCTs and observational studies. And I am not sure that either of these more novel aspects move the field forward appreciably.

More specific comments: While it appears the aims of the Start Up program may have been primarily convenience (particularly for clinicians) and support (particularly for patients), from a research perspective a few more details would be helpful. Were there any prespecified hypotheses or aims? If so, would add to abstract and methods. Even if not, consider adding as the last sentence of the introduction some sort of general aim or even just a “what this paper is about” sentence. Something like “In the current report, we describe…”

For AE assessing / grading / reporting, as outlined in the methods, more detail is needed. During the phone calls, were patients asked only general question (how are you feeling / any problems) and then any spontaneous AEs were graded? Or were there specific queries focused on the most common prostanoid AEs such as GI side effects or headache? And was any sort of script utilized? Did the nurse who was on the phone grade the AEs, or was this done by a physician or researcher?

For the discontinuation rates (for reasons other than death) – how does this compare across RCT and observational studies?

What medications were utilized to treat AEs, if any (medications for headache, nausea, diarrhea, for example). Were these recommended by the nurse? Was there any sort of preplanned AE management in place? Consider a table with information on med use.

Minor points

- In the results section, avoid using terms such as “benefited” (patients “benefitted” from a phone call) and instead use more neutral terms such as received.

- When referring to physicians on line 162, favor rewording to avoid the use of “him” in this context; perhaps “A systemic report was provided to the treating physicians, providing insight…”

- Line 164 and 166, add s to patient

- Line 267, add “of” before patients

The patient satisfaction questionnaire completion rate (28%) is quite low – any comments? And what was the completion rate amongst health care providers?

**Reviewer #2: **Thanks for the opportunity to review this paper re. follow-up a medication where side-effects often hinder optimal dosage. Some comments as they appear in the text.

Page 2, line 27, write out the CTCAE v4.0.

Page 5, line 71, write out HTaPFrance.

Page 5, the timeline described in line 76-81 could be more easily described if Figure 1 is referenced. I may read badly, but the numbers do not fit with the description in results from line 146 onwards.

Page 8, line 143, what is MR004 reference methodology (no reference given).

Page 12, line 184-186, I miss the definition of high, medium, low in the Methods section. Were these cut-offs decided a priori?

Page 16, “alert from the French drug regulatory agency”, please explain (not only in the footnote to Figure 1).

Page 16, line 238, very few patients answered the rating, please discuss if the findings are representative with such a low response rate. Same page, what was the response rate amongst health professionals? LATER, I see the discussion under limitations on page 21. But the authors could speculate more any effect of biases.

Figure 2, please give a explanatory footnote. As now, the figure is not understandable.

Figure 4, what number is correct? In text, out of 302 reaching dose, 86 answered. In Figure 4, 249 questionnaires were sent. Please check and be careful with numbers and be consistent!

6. PLOS authors have the option to publish the peer review history of their article (what does this mean?). If published, this will include your full peer review and any attached files.

Reviewer #1: No

Reviewer #2: **Yes: **Stefan Söderberg

---

## [Author Response · Author response to Decision Letter 1]

19 Jun 2025

Dear PlosOne editor, dear reviewers,

We would like to thank you for these constructive comments on our work. Please find below our responses to all of your concerns.

Academic Editor comments:

Style requirements: Requirements regarding style and file naming are now met in the updated version.

Funding statement: All funding is declared in the funding statement. The statement “There was no additional external funding received for this study” has been added to the reviewed cover letter.

Data accessibility: Data are issued from the Start Up patient support program developed by Patientys. Patient gave their written consent for the use of their data, however they did not agree specifically for public availability of their unaggregated data. To meet PlosOne requirements, data can be available upon request at privacy@patientys.com. The section has been updated in the submission form.

Ethics statement: Our ethics statement has been updated in the methods section with complete information about the institutional body.

Reference list: The reference list has been checked to ensure that it is complete and correct.

Reviewer#1 comments

Objective of the manuscript: The Start Up program was first designed to bring support to patients and their medical teams while initiating prostacyclin. The objective of the current study was to report the existence of such patient support program and give real world data supporting the hypothesis that an external support is beneficial to patients and physicians in treatment titration. Our data shows that the external nurse allows for a personalized titration pattern following AE reported by patients which may participate to a higher rate of patient reaching their optimal dose and a lower discontinuation rate. In addition, the time spent by the external nurse is time saving for medical teams and highly appreciated.

AE assessing: During the phone calls, using guidelines determined by the scientific committee, the nurse drives the patient interview to identify and grade AEs. There is a specific focus on prostanoids associated AEs, in order to give insights to the medical team of the patient that will take decision regarding the treatment. The methods section has been detailed.

Medications use: Medications to treat AEs were only manage by the medical team of the patient, in regard to the AEs declared by the patient. Some medical teams anticipated and gave to their patient a prescription for medication to treat AEs while introducing Selexipag. The PAH nurse of the Start Up program may solicit the physician to send a prescription if needed but do not take any medical decisions. The methods section has been detailed.

Discontinuation rates compared to current literature In the Start Up program, 78 patients (19%) discontinued their treatment, before reaching their optimal treatment dose. Among them, 35 patients (9%) reported a treatment discontinuation due to adverse events during the titration phase.

In the Phase 3 clinical trial (GRIPHON), 14,3% of patient reported a treatment discontinuation related to adverse event. Among the several real-world data studies, more relatable to the Start Up program, the discontinuation rate due to adverse events varies from 20% in the EXPOSURE study and 22% in the SPHERE registry study up to 32% in the GRIPHON open label study. However, they monitor discontinuation after the titration period, making a comparison more difficult with our study. We could only speculate that because adverse events occur more frequently during the dose-adjustment phase (Sitbon et al., 2015), treatment discontinuation may have been higher during the titration phase.

Therefore, the discontinuation rate observed in the Start Up study may be lower than other real world data studies, supporting that the close patient monitoring may participate to less treatment discontinuation. � The discussion section has been detailed.

Minor points:

- “Benefited” � “Received”: modified

- Avoid the use of “him” when referring to physician: modified

- Line 164 and 166: Add “S” to patient: modified

- Line 267: ass “of”: modified

Patients’ satisfaction completion: The satisfaction questionnaire was only implemented later in 2018, as described in the Methods section and sent to 249 patients out of the 302 patients that reached their optimal dose. In addition, patients must fill-out the satisfaction questionnaire and send it using a prepaid envelope to the PAH nurse. This postal sent by the patient may explain the low completion rate.

Completion rate of health care providers: Unfortunately, amongst healthcare providers, the distribution of the questionnaires was not controlled. While the questionnaire was sent by email to 64 physicians, we received answers from nurses and pharmacists suggesting email forwarding. We cannot therefore provide a response rate among healthcare professionals.

Reviewer#2 comments

Page 2, line 27, write out the CTCAE v4.0. � detailed

Page 5, line 71, write out HTaPFrance. � detailed

Page 5, the timeline described in line 76-81 could be more easily described if Figure 1 is referenced. I may read badly, but the numbers do not fit with the description in results from line 146 onwards. � Line 76-81 aimed to describe the 2 populations of patients that benefited or not of the additional follow-up of 1 year. Line 149 and followings describe the whole population and detail the patients that did not reached their individualized dose. For a better clarity the Methods section is detailed and figure 1 referenced.

Page 8, line 143, what is MR004 reference methodology (no reference given). � detailed

Page 12, line 184-186, I miss the definition of high, medium, low in the Methods section. Were these cut-offs decided a priori? � This is aligned to the GRIPHON study, Phase III trial for Selexipag. The Methods section has been detailed.

Page 16, “alert from the French drug regulatory agency”, please explain (not only in the footnote to Figure 1). � Explained in the Results section

Page 16, line 238, very few patients answered the rating, please discuss if the findings are representative with such a low response rate. Same page, what was the response rate amongst health professionals? LATER, I see the discussion under limitations on page 21. But the authors could speculate more any effect of biases. � Discussion more detailed. The low response rate among patients may be due to the data collection method for satisfaction. Patients had to return the questionnaire in a pre-stamped envelope. Unfortunately, amongst healthcare providers, the distribution of the questionnaires was not controlled. While the questionnaire was sent by email to 64 physicians, we received answers from nurses and pharmacists suggesting email forwarding. We cannot therefore provide a response rate among healthcare professionals.

Figure 2, please give an explanatory footnote. As now, the figure is not understandable. � Detailed

Figure 4, what number is correct? In text, out of 302 reaching dose, 86 answered. In Figure 4, 249 questionnaires were sent. Please check and be careful with numbers and be consistent! � Of the 302 patients that reached their optimal dose, the questionnaire was sent to 249 patients. This is due to the implementation of the satisfaction questionnaire later in the program resulting in early participants not filling the questionnaire. The results section has been detailed to be consistent.

---

## [Editor Report · Decision Letter 1]

24 Jun 2025

PONE-D-25-06416R1Start Up: A French program to support patients with pulmonary arterial hypertension during the adjustment of prostanoids to the individualized optimal dosePLOS ONE

Dear Dr. Bergot,

Thank you for submitting your manuscript to PLOS ONE. After careful consideration, we feel that it has merit but does not fully meet PLOS ONE’s publication criteria as it currently stands. Therefore, we invite you to submit a revised version of the manuscript that addresses the points raised during the review process.

We look forward to receiving your revised manuscript.

Kind regards,

James P. Maloney, MD

Academic Editor

PLOS ONE

Journal Requirements:

Editor Comments:

The manuscript provides valuable information on selexipag titration success rates, and the manuscript has improved after changes made in response to reviewers. However, several areas of the reviewer comments still need to be addressed, mostly related to industry-driven bias for the project and related statements in the manuscript. As the titration protocol appears proprietary, the implications of the project outside of France are unclear. The authors need to clarify what aspects of this program are proprietary and minimize industry bias.

1. Reviewer 1 asked for clarification of the purpose of the program, and for any hypotheses that accompanied this project. While the revision does not mention hypotheses, it does clarify intent - "objective to support them in taking their therapies targeting the prostacyclin pathway and determine the individualized optimal dose". However, the project is sponsored by two for profit companies, one the drug manufacturer, so the manuscript needs to clarify the role of these companies for this project in multiple sections (not just in the Funding statement).

a) Is "Start Up" name copyrighted or is that being sought ? (or its equivalent in the EU). Is the name owned by someone?

b) The title refers to this as a "French" program, and indeed Patientys and Janssen-Cilag are in France, but both are for-profit and Janssen is world-wide and publicly traded. Is there an intent to market this program outside of France? The manuscript says the program was developed by the French PH network - was it then sold or licensed to Patientys? Was Patientys involved in the genesis of the original version of the program? Who pays for the nurse coordinator in this project - Janssen or Patientys? The Patientys website is only in French, so the authors need to clarify more aspects of this company and if it is in part owned by Janssen.

c) The term "external" is misleading (line 72), as the "external" nurse is employed by a for-profit company. Thus, this should be clarified to "an external nurse employed by Patientys, directly supported by Janssen." State if the nurse was a patientys or Janssen employee. The role and for-profit nature of the companies must be clear in the Methods section.

d) the StartUp program thus appears proprietary and is owned/co-owned by two for-profit companies. Make clear that the details of the program/standard operating procedures (SOP)/counseling/etc are proprietary and are not part of the publication. The structure of the program is of interest to readers, if it can't be released then state so - that it is proprietary. If it can be released, add it as an appendix.

e) Line 23, what does "supported by the French PH network" mean? Financially supported? Did the French PH network use its own nurses then, or were patientys/janssen involved from the start? If not, when did the transition occur?

d) As 2 authors are from Janssen (AT and TB), then Janssen should appear in the text of the manuscript, not just in author and funding sections. Make it clear that Janssen is the drug manufacturer in the methods section.

e) the weblink to the Janssen website is not relevant or helpful to the reader (it does not even go directly to selexipag information), thus remove it

f) This program appears similar to programs such as BAYER that were designed to support riociguat use in the USA, so the program is not novel. Are there publications on the BAYER program for comparison? In the USA the FDA mandated nurse-assessed BP and side effect evaluations during riociguat titration. However, in the USA most, but not all commercial payers will cover the extra cost of the RN monitoring program during drug uptitration, while some insurers will not pay for the extra charges for the nurse-driven program. Selexipag does not have such a program in the USA, in USA PH centers the nurses and providers steer uptitration, without a defined SOP. How does this work in France? Outside of the StartUp program, how is selexipag uptitrated in France? Is the drug only uptitrated via StartUp?   Including a discussion of other PH drug titration patient assistance programs, base don publicly available information, will be very helpful for context of the StartUp program.

g): Getting to reviewer 1's comments, the authors now add more discussion of other selexipag trials and compare the apparently lower selexipag discontinuation rates. However, the authors need to add to the limitations paragraph a discussion that direct statistical comparison was not performed.

h) correct BIAS: line 307-8, since no references are given, defend the statement included or remove it: "In contrast with existing PAH patient support programs, Start Up was supervised by a scientific medical committee". Does this team know that BAYER did not have a "scientific medical committee" for its program? As above, clarify when Patientys and Janssen came into the project - if they funded the French PH network from the start, say so. If the program was transitioned to Patientys at some stage, clarify that. Line 308 - the "scientific medical committee" makeup needs to be stated (even as an appendix, so the reader can ascertain the balance of French university-hospital based members versus industry members).

2) Why is data after 2022 not included? Did the program end - was it a pilot?

3) Line 175: the authors still do not clarify the nature of the selexipag patient alert requested by reviewers, and appear to purposely leave it out. Giving the website link is of little help as it is only in French. State the reason for the alert very specifically, even if Janssen co-authors view it adversely.

4) Line 254: BIAS - remove the position statement that the alert was "unsubstantiated". The reader cannot verify this statement, unless the French public oversight agency later retracted the alert and themselves used that word.

5) Strike much of line 337 as a cost analysis was not performed, so the authors cannot say the program is "profitable" which implies finances. The authors can only state "useful" in limiting drug discontinuation rates compared to RCTs, and base don survey data.

Minor Comments

- line 392: "representation" not "representability"

- line 391: awkward, correct

- line 404: their, not "her"

---

## [Author Response · Author response to Decision Letter 2]

25 Jul 2025

In response to editor's comments and requests on the manuscript the Response to reviewers has been updated.

We clarified the specific status of the Start Up program in France which falls in the French regulation framework and the role of the for-profit organisations and non-profit organisation. The Start Up program has been initiated by the non profit organisation, constructed by a scientific committee, and implemented by Patientys, which acts as a trusted third party and operational organisation. Janssen is the institutionnal support of the project, allowing for patient and hospital to benefit from the PSP free of charge. The drug cost management is independant of the program as it is covered by the French healthcare system.

I have downloaded an updated cover letter, manuscript and response to reviewers.

Regards,

---

## [Editor Report · Decision Letter 2]

30 Jul 2025

PONE-D-25-06416R2Start Up: A French program to support patients with pulmonary arterial hypertension during the adjustment of prostanoids to the individualized optimal dosePLOS ONE

Dear Dr. Bergot,

Thank you for submitting your manuscript to PLOS ONE. After careful consideration, we feel that it has merit but does not fully meet PLOS ONE’s publication criteria as it currently stands. Therefore, we invite you to submit a revised (minor) version of the manuscript that addresses the points raised during the review process.

We look forward to receiving your revised manuscript.

Kind regards,

James P. Maloney, MD

Academic Editor

PLOS ONE

Journal Requirements:

Additional Editor Comments:

This revision is a better manuscript but it has new typo and grammar errors in sections the authors edited. These need to be fixed.

1. Line 339: "may represent", no need for S on the end

2. Line 393: should read "did not benefit"

3. Line 393: Should read "Nor is there a direct..."

4. line 503: representation

5. line 398: "suggest", not suppose

6. Line 399: should read "...the overall population..."

7. Line 402: should read "limited our rate of response....", remove 'satisfaction"

---

## [Author Response · Author response to Decision Letter 3]

5 Aug 2025

Dear Plos One Editor,

Typo and grammar errors have been taken into account in the revised manuscript. The Manuscript, Manuscript with track changes and Responses to reviewers are uploaded.

Kind regards,

---

## [Editor Report · Decision Letter 3]

10 Aug 2025

Start Up: A French program to support patients with pulmonary arterial hypertension during the adjustment of prostanoids to the individualized optimal dose

PONE-D-25-06416R3

Dear Dr. Bergot,

We’re pleased to inform you that your manuscript has been judged scientifically suitable for publication and will be formally accepted for publication once it meets all outstanding technical requirements.

Kind regards,

James P. Maloney, MD

Academic Editor

PLOS ONE

Additional Editor Comments (optional):

Grammar and typographical errors are now fixed.
---

## [Editor Report · Acceptance letter]

PONE-D-25-06416R3

PLOS ONE

Dear Dr. Bergot,

I'm pleased to inform you that your manuscript has been deemed suitable for publication in PLOS ONE. Congratulations! Your manuscript is now being handed over to our production team.

Kind regards,

on behalf of

Dr. James P. Maloney

Academic Editor

PLOS ONE